# The Impact of Maternal Probiotics on Intestinal Vitamin D Receptor Expression in Early Life

**DOI:** 10.3390/biom13050847

**Published:** 2023-05-16

**Authors:** Anita Sharma, Yueyue Yu, Jing Lu, Lei Lu, Yong-Guo Zhang, Yinglin Xia, Jun Sun, Erika C. Claud

**Affiliations:** 1Division of Pediatric Gastroenterology, C.S. Mott Children’s Hospital, Michigan Medicine, University of Michigan, Ann Arbor, MI 48109, USA; 2Department of Pediatrics, University of Chicago, Chicago, IL 60637, USA; 3Division of Gastroenterology and Hepatology, Department of Medicine, University of Illinois at Chicago, Chicago, IL 60607, USA

**Keywords:** vitamin D, VDR, neonatal, infant, intestine, inflammation, NF-κB, probiotic, microbiome

## Abstract

Vitamin D signaling via the Vitamin D Receptor (VDR) has been shown to protect against intestinal inflammation. Previous studies have also reported the mutual interactions of intestinal VDR and the microbiome, indicating a potential role of probiotics in modulating VDR expression. In preterm infants, although probiotics have been shown to reduce the incidence of necrotizing enterocolitis (NEC), they are not currently recommended by the FDA due to potential risks in this population. No previous studies have delved into the effect of maternally administered probiotics on intestinal VDR expression in early life. Using an infancy mouse model, we found that young mice exposed to maternally administered probiotics (SPF/LB) maintained higher colonic VDR expression than our unexposed mice (SPF) in the face of a systemic inflammatory stimulus. These findings indicate a potential role for microbiome-modulating therapies in preventing diseases such as NEC through the enhancement of VDR signaling.

## 1. Introduction

Vitamin D deficiency has relevance beyond bone health and is associated with multiple prematurity complications such as bronchopulmonary dysplasia, necrotizing enterocolitis, and neonatal sepsis [1,2,3,4,5]. Vitamin D signaling involves the binding of Calcitriol (1,25(OH)_2_D) to the Vitamin D Receptor (VDR) in target tissues. VDR is a member of the nuclear receptor superfamily and is highly expressed in the intestines, functioning as a transcription factor. More than 50 target genes of VDR have been identified that have an important role in the innate immune system and the intestinal microbiome [6]. VDR has been found to be intestinally protective and anti-inflammatory in numerous studies [7,8,9,10,11,12,13,14,15,16], and cells lacking VDR are more likely to be in a proinflammatory state with the activation of NF-κB [17].

Interestingly, probiotics have been shown to increase the expression of intestinal VDR at the transcription level in times of infection and inflammation [8,18]. Furthermore, probiotics decrease the incidence of necrotizing enterocolitis (NEC) [19,20,21,22], which is an inflammatory bowel necrosis that primarily afflicts preterm infants after the onset of enteral feeding. Specifically, the probiotic c strains *Lactobacillus acidophilus* and *Bifidobacterium infantis* (LB) are well documented in the prevention of NEC [23]. However, the effects of probiotics on Vitamin D/VDR signaling in the immature intestine have not been previously explored. Furthermore, concerns regarding the safety of probiotics in the immunocompromised premature neonate have limited routine clinical use in preterm infants [24]. 

We hypothesized that the probiotic compound LB given to pregnant mothers from late pregnancy through lactation may be a means of optimizing the infant gut microbiome to influence Vitamin D status as well as the expression of colonic VDR and NF-κB at baseline, and in the setting of a systemic inflammatory stimulus via administration of intraperitoneal (i.p.) IL-1β, a proinflammatory cytokine that has been shown to increase intestinal permeability [25]. Pre-weaned (2-weeks-old) and post-weaned (4-weeks-old) pups were used for this study to model preterm and term neonates, respectively, as pre-weaned mouse intestinal epithelium (P14) is developmentally comparable to preterm infant intestine [26].

## 2. Materials and Methods

### 2.1. Animals

This study was carried out in strict accordance with the recommendations in the Guidelines for the Care and Use of Laboratory Animals from the National Institutes of Health. All animal work was conducted under animal protocol No. 71703 and was approved by the University of Chicago Institutional Animal Care and Use Committee (IACUC). Time-dated pregnant C57/BL6J specific pathogen-free (SPF) mice were obtained from Jackson Laboratory (Bar Harbor, ME) and were kept on a 12 h light/dark cycle with access to food and water ad libitum. On embryonic day 15 (ED15), SPF dams were fed a 10^9^ colony forming unit (cfu) dosage each of *Lactobacillus acidophilus* and *Bifidobacterium infantis* (LB) daily until pup sacrifice, and the offspring are denoted throughout the results as SPF/LB. Control SPF-pregnant dams were fed PBS and the offspring are denoted throughout the results as SPF. No side effects such as diarrhea were observed in the dams who received LB supplementation. The dams used in this study were about 3 months old and the average dam weight was about 35 g. There were approximately 6–8 pups per litter. Pups were delivered naturally and remained with their mothers until sacrifice. Pre-weaned (2-weeks-old) and post-weaned (4-weeks-old) pups were used for this study, as pre-weaned mouse intestinal epithelium (P14) is developmentally comparable to the preterm infant intestine [26]. Besides routine monitoring of animal well-being at the facility, during the experimental periods, the dams were checked and weighed daily by a trained technician. If distress signs such as hunched back, weight loss, dystocia or lethargy appeared, the animals would be euthanized per institutional guidelines using CO_2_ followed by cervical dislocation.

The pups were divided in two groups and were either intraperitoneal (i.p.) injected with saline as a control or i.p. injected with IL-1β at 50 ng/g body weight to model the systemic inflammation observed in NEC patients. After four hours of IL-1β (cat# 50813285, Thermo-Fisher, Waltham, MA, USA) treatment, mice were euthanized under isoflurane followed by cardiac puncture for blood collection, and sera were collected and stored at −80 °C for further analysis. Colonic tissue was harvested and snap frozen on a dry ice-ethanol bath for biochemical analysis, and fixed and processed for histological and immunohistochemical analysis.

### 2.2. Probiotic Bacteria Preparation

*Lactobacillus acidophilus*, and *Bifidobacteria infantis* used were from ATCC (No. 53544 and 15697, respectively). Bacteria were first grown and expanded in MRS broth (DeMan Rogosa & Sharpe, Difco, Fisher Scientific, Loughborough, UK) at 37 °C and 5% CO_2_ under anaerobic and non-agitating conditions, centrifuged (20 min, 4500× *g*), and resuspended in modified Hank’s balanced saline solution (HBSS) supplemented with 0.04 M MgSO_4_, 0.03 M MnSO_4_, 1.15 M K_2_PO_4_, 0.36 M sodium acetate, 0.88 M ammonium citrate, 10% polysorbate (growth factor for *Lactobacillus* sp) and 20% dextrose. Bacteria were then propagated overnight at 37 °C, 5% CO_2_ under non-agitating conditions to 2 × 10^9^ cfu/mL.

### 2.3. ELISA

ELISA was performed on mouse sera for 2- and 4-week-old mice using the Mouse Rat 25-OH Vitamin D ELISA kit from Eagle Biosciences (SKU: VID21-K01) as per manufacturer’s instructions.

### 2.4. Immunohistochemistry

Colonic tissue was obtained from pre-weaned and post-weaned mice and was formalin-fixed, paraffin-embedded. Sections (5 μm) were cut and used for morphology and immunohistology.

For Immunohistochemistry and immunofluorescence studies, colonic sections from SPF and SPF+LB groups were deparaffined, rehydrated, and then incubated with blocking solution (5% goat serum and 2% BSA) in TBST. The tissue sections were incubated with 50 μL of the respective primary antibody (VDR (1:200, Santa Cruz, SC-13133) and phospho-p65 (1:200, Abcam, ab86299)) overnight at 4 °C. After washing with TBST, the sections were incubated with the respective fluorophore-conjugated secondary antibodies and then counterstained with DAPI-antifade mounting medium (P36935, Invitrogen Inc., Carlsbad, CA, USA). Imaging was performed at the University of Chicago Integrated Light Microscopy Facility. Imaging, processing, and analysis were obtained using QuPath software measuring the percentage of DAB-positive cells.

### 2.5. Statistics

The results from the above data are presented as mean ± SEM. Student’s *t* test was performed for two group comparisons using GraphPad Prism version 6.00 for Windows (GraphPad Software, La Jolla, CA, USA, www.graphpad.com accessed on 15 January 2021). A *p*-value of less than 0.05 was considered statistically significant.

## 3. Results

Since Vitamin D is important in infant health, we first determined the impact of probiotics on serum Vitamin D levels in young mice. At 2 weeks of age, probiotic-exposed mice (SPF/LB) maintained significantly higher serum 25(OH) Vitamin D levels than unexposed SPF mice (110.2 ng/mL in SPF vs. 135.5 ng/mL in SPF+LB, *p* = 0.04). At 4 weeks of age, when the mice were no longer co-housed with their mothers, there was no significant difference in 25(OH) Vitamin D levels between SPF and SPF/LB mice (73.10 ng/mL in SPF vs. 71.30 ng/mL in SPF+LB, *p* = 0.60) (Figure 1).

As Vitamin D signaling is dependent on VDR at the tissue level, we next determined the impact of probiotics on VDR expression in the colon. At 2 weeks of age, the probiotic-exposed subgroup SPF/LB was found to have significantly higher colonic VDR expression at baseline compared to SPF mice (36.96% and 25.11%, respectively; *p* = 0.01) (Figure 2a). This, however, was not seen at 4 weeks of age, as VDR expression was comparable between the SPF and SPF/LB subgroups (29.15% and 34.62%, respectively; *p* = 0.09) (Figure 2b).

Given that VDR has known cytoprotective/immunomodulatory roles, and that previous studies have shown that VDR expression may be downregulated in inflammatory states [27], we next determined how VDR expression responds to systemic inflammation at both time points. In the pre-weaning period, there was no significant change in VDR expression with exposure to IL-1β in either SPF or SPF/LB mice (Appendix A). In post-weaned 4-week-old mice, systemic inflammation downregulated the colonic VDR expression in the SPF subgroup (29.15% in SPF and 14.06% SPF treated with IL-1β, *p* < 0.0001) but not in the SPF/LB mice, who maintained VDR expression with no significant downregulation upon exposure to IL-1β (34.62% in SPF/LB and 30.13% SPF/LB treated with IL-1β, *p* = 0.13) (Figure 3).

To further clarify the anti-inflammatory role of VDR, we next explored the relationship between VDR expression and expression of the phosphorylated p65 subunit of NF-κB (phospho-p65). p65 is a protein involved in NF-κB heterodimer formation, nuclear translocation, and activation. NF-κB signaling may be activated upon exposure to a systemic inflammatory stimulus such as IL-1β [28], and we thus hypothesized that phospho-p65 expression would be increased in IL-1β-exposed mice and would be inversely associated with VDR. Notably, at 4 weeks of age, baseline phospho-p65 expression was significantly higher in SPF than in SPF/LB mice (*p* = 0.01). We found that upon exposure to IL-1β, both the SPF and SPF/LB mice had upregulation of colonic phospho-p65 expression (7.79% in SPF and 33.16% in SPF with IL-1β, *p* < 0.0001; 2.56% in SPF/LB and 7.92% in SPF/LB with IL-1β, *p* = 0.0003). However, the increase in p65 with IL-1β treatment was significantly more substantial in our SPF mice compared to our SPF/LB mice (33.16% and 7.92%, respectively; *p* = 0.0007) (Figure 4). 

At 2 weeks of age, there was no significant difference in phospho-p65 expression with systemic IL-1β in either SPF or SPF/LB mice, which corresponds to the lack of difference seen in VDR expression following IL-1β treatment (Appendix A).

## 4. Discussion

Our study shows that maternally administered probiotics impact offspring biology and are protective in the vulnerable neonatal period through early maturation and upregulation of colonic VDR, which is associated with decreased inflammation through the NF-κB pathway.

Vitamin D status denoted by serum 25(OH) Vitamin D levels is considered to be a marker for optimal Vitamin D signaling [29]. At 2 weeks of age, which is the time of relevance for preterm infants, SPF+LB mice had significantly higher 25(OH) Vitamin D levels than SPF mice, and this correlated with their colonic VDR expression. At 4 weeks of age, SPF and SPF+LB mice had no significant difference in their serum 25(OH) Vitamin D levels, which also reflected the pattern seen in their colonic VDR expression. However, probiotics conferred increased intestinal protection via upregulation of colonic VDR in response to inflammation. Therefore, probiotics may play a more direct role at the tissue-level in optimizing Vitamin D signaling through its receptor VDR, and this may not always be captured by studying serum 25(OH) vitamin D levels alone.

Interestingly, we found that pre-weaned SPF mice maintained colonic VDR and had no significant upregulation of phospho-p65 expression with a systemic inflammatory stimulus, which was a pattern not seen in the post-weaned SPF mice. We surmise this may be due to added protection from their mothers during the preweaning/cohousing period.

Pre-weaned mouse pups also maintained higher serum 25(OH) Vitamin D levels overall compared to their post-weaned counterparts; we can only hypothesize that this may be due to the effects of mother’s milk and cohousing.

There are some study limitations that must be considered. For one, we do not have data on serum Vitamin D levels or the gut VDR expression of the dams involved. The study sample sizes were also small. Additionally, this study focuses on qualitative VDR protein expression differences, and we do not have comparable quantitative data (such as through PCR), given the limitations of PCR measurements of intestinal VDR.

This is the first study to demonstrate that probiotics given to mothers during the prenatal period may effectively influence infant Vitamin D status and decrease inflammation by upregulating colonic VDR. This is of high significance because of the importance of vitamin D to health outcomes and the safety concerns regarding administering probiotics directly to infants. Our previous studies in adult models have demonstrated that VDR negatively regulates the activity of the NF-κB pathway by physically binding with NF-κB p65 and transcriptionally modulating IκBα [17,30]. In the current study, we observed the reduction of VDR after IL-1β treatment. Furthermore, 1,25D3 is known to boosts infection-stimulated cytokine/chemokines, a critical component of the macrophage response to *Mycobacteria marinum* infection [31], and 1,25D directly upregulates IL-1β gene transcription in macrophages. Although the mutual interactions between intestinal vitamin D/VDR and IL-1β are not well established, Vitamin D/VDR can modulate the innate and adaptive immune responses [32]. Thus, future studies are needed to further delineate VDR signaling in the colon and offer mechanistic insights into the relationship between VDR and NF-κB at the tissue level. Our study of Vitamin D and VDR in preterm infants with a focus on intestinal biology and inflammatory signaling may be critical for mitigating morbidities and improving preterm infant outcomes.

## Figures and Tables

**Figure 1 biomolecules-13-00847-f001:**
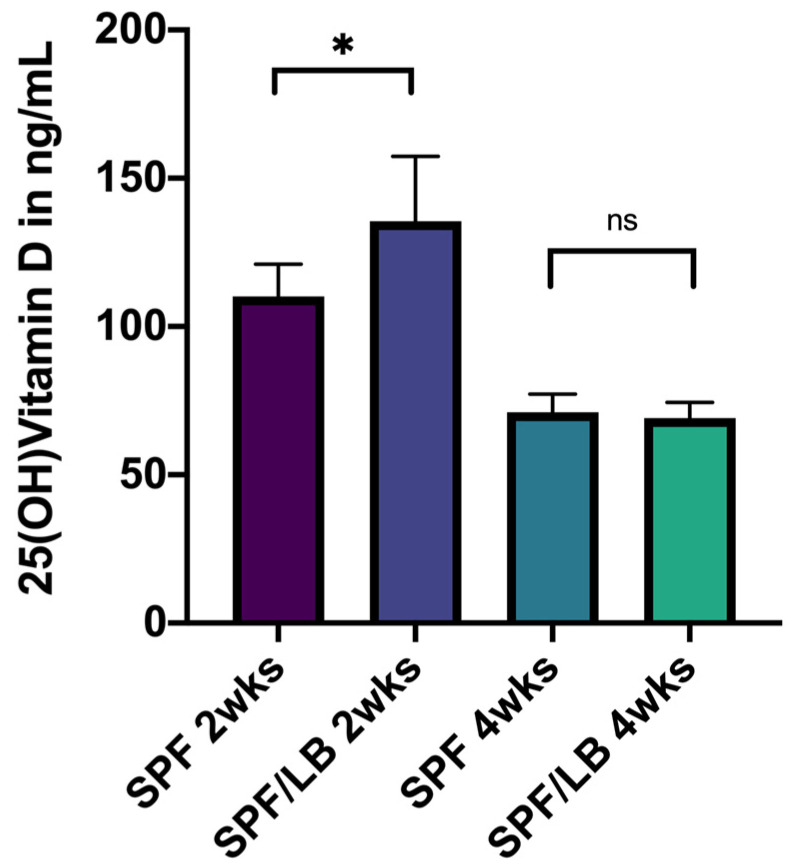
25(OH) Vitamin D levels in 2-week-old and 4-week-old SPF and SPF/LB mice. * denotes significance with *p*-value = 0.04; ns = non significance with *p*-value 0.60; *n* ≥ 3 for all subgroups.

**Figure 2 biomolecules-13-00847-f002:**
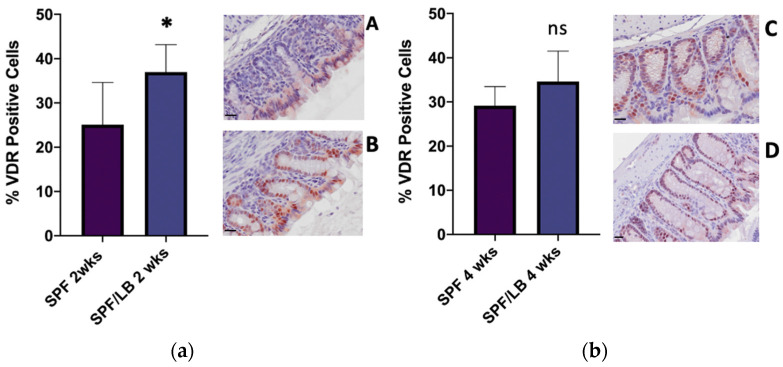
(**a**) baseline colonic VDR expression and representative immunohistochemistry in: 2-week-old SPF (**A**) and SPF/LB (**B**) mice. * denotes significance with *p*-value = 0.01. There were 5 pups included in SPF and 3 pups included in SPF/LB subgroups. Scale bar = 20 μm; (**b**) baseline colonic VDR expression and representative immunohistochemistry in 4-week-old SPF (**C**) and SPF/LB (**D**) mice. ns = non-significance with *p*-value 0.19. 3 pups were included in SPF and 4 pups were included in SPF/LB subgroups. Scale bar = 20 μm.

**Figure 3 biomolecules-13-00847-f003:**
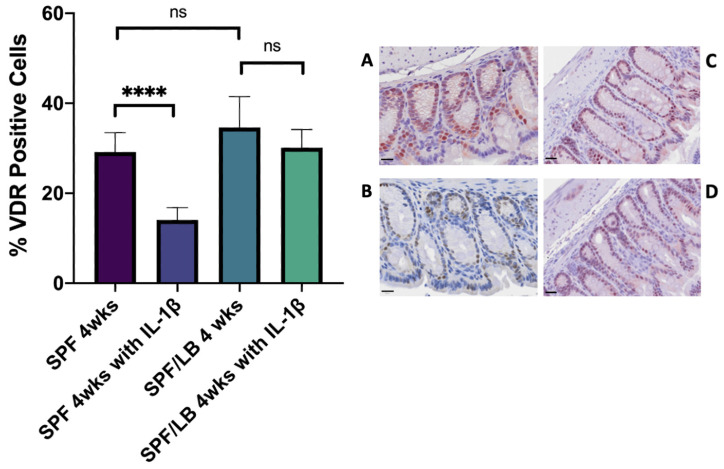
Colonic VDR expression and representative immunohistochemistry in: 4-week-old SPF mice at baseline (**A**) and following i.p. treatment with IL-1β (**B**). **** denotes statistical significance with *p*-value < 0.0001. There were 3 pups included in SPF and 6 pups included in SPF with IL-1β subgroups. Scale bar = 20 μm; 4-week-old SPF/LB mice at baseline (**C**) and following i.p. treatment with IL-1β (**D**); ns denotes non-significance with *p*-value 0.13. There were 4 pups included in SPF/LB and 5 pups included in SPF/LB with IL-1β subgroups. Scale bar = 20 μm.

**Figure 4 biomolecules-13-00847-f004:**
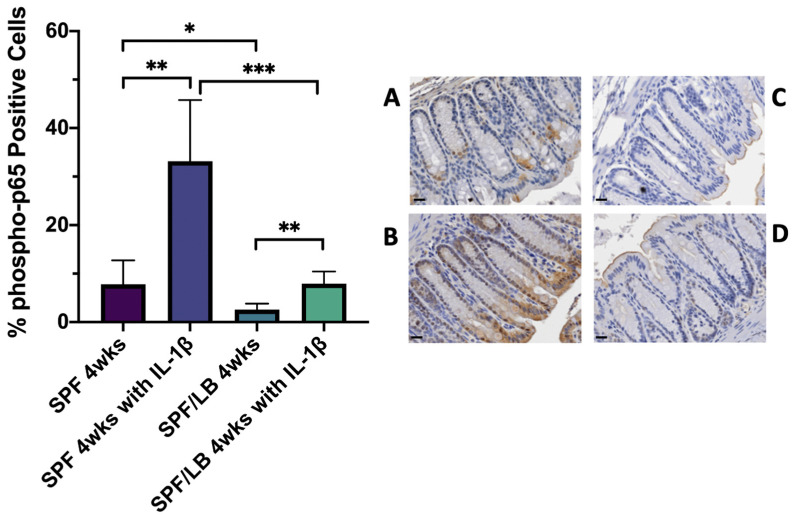
Colonic phospho-p65 expression and representative immunohistochemistry in: 4-week-old SPF (**A**) and SPF with IL-1β (**B**) mice. ** denotes statistical significance with *p*-value = 0.003. There were 3 pups included in SPF and 6 pups included in SPF with IL-1β subgroups. Scale bar = 20 μm; 4-week-old SPF/LB (**C**) and SPF/LB with IL-1β (**D**) mice. ** denotes statistical significance with *p*-value = 0.004. There were 4 pups included in SPF/LB and 5 pups included in SPF/LB with IL-1β subgroups. Scale bar = 20 μm; further subgroup analysis compared: 4-week-old SPF and SPF/LB mice; * denotes statistical significance with *p*-value = 0.01; 4-week-old SPF with IL-1β and SPF/LB with IL-1β mice; *** denotes statistical significance with *p*-value = 0.0007.

## Data Availability

Data are contained within the article or Appendix A.

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
