# Peer review of "The Impact of Maternal Probiotics on Intestinal Vitamin D Receptor Expression in Early Life"

_biomolecules, 2023, doi:10.3390/biom13050847_

Round 1

Reviewer 1 Report

Dear Editor,

This manuscript reads well, and the experiments were well executed. The authors showed modulation of vitamin D receptor (VDR) expression in the gut of offspring whose mothers received probiotic supplementation during gestation. It is an interesting and original finding; however, I missed the mechanistic explanation for this from further experiments and/or from the available literature (in the discussion session). Furthermore, it was unclear to me why the authors decided to investigate the VD levels and VDR in this experimental model of “fetal programming”. This should be better explained in the rationale of the study.

Although the experiments were well conducted, the authors could have explored in more depth their main findings. For instance, experiments to assess gene expression of VDR would add internal validity to their major results. I would also suggest the authors to report the serum VD levels and the gut VDR expression of the dams. Would probiotic supplementations have similar effect in dams and offspring?

Additional comments:

-       Authors should report:

o   The sample size for each experiment.

o   Age/body weight of the dams used and if there was any difference between groups.

o   If side effects, e.g. diarrhoea, was observed during probiotic supplementation.

o   Litter size, still birth, and anthropometric measurements of the offspring.

-       Authors should explain the significant drop in serum VD levels that occurred in mice from 2 to 4 weeks of age.

-       Supplemental figure should be included in the main manuscript if the editorial office is happy with this.

Reviewer 2 Report

The com.smunication article by Sharma et al., shoed the impact of maternal probiotics on intestinal Vitamin D receptor expression in early life. This study is very impactful as vitamin D receptor plays key  role in maintaining immunological homeostasis. However, there are some comments that will further improve the manuscripts.

  • Relationship between vdr and il1b is not clear. how vit d affect il1b response through vdg? Please elaborately discuss this point. 
  • Represent the data as scatter dot plot
  • atleast another molecular biology experiment like real time pcr data is needed to confirm the ihc findings
  • Please discuss the following articles in discussion (PMID : 36562589, 26159695, 32170938)
